# [Regular Track] AgentTravel: Knowledge-Augmented LLM Agent Framework for Urban Travel Planning

**Jie Zhao, Jie Feng, Yong Li**[*]

Department of Electronic Engineering, Tsinghua University, Beijing, China,
Beijing National Research Center for Information Science and Technology (BNRist), China
`csjiezhao@gmail.com`, `{fengjie,liyong07}@tsinghua.edu.cn`

## Abstract

Large language models are opening new opportunities for intelligent decision support, with urban travel planning as a challenging and high-impact use case. Effective planning requires integrating real-time, multi-source data—such as points of interest, transportation, and user preferences—while reasoning spatially to generate feasible itineraries. This paper proposes AgentTravel, a unified framework that combines knowledge-grounded modeling, agentic reasoning, and multi-perspective evaluation. It includes: 1) TravelLLM, a domain-adapted model enriched with urban and spatial knowledge; 2) TravelAgent, an agentic planner with structured itinerary memory and real-time data retrieval; and 3) TravelBench, a benchmark assessing both knowledge grounding and plan quality. Experiments on five Chinese cities show that AgentTravel surpasses strong baselines in factual reasoning and itinerary feasibility, offering a promising step toward grounded and adaptive LLMs for urban intelligence. Source code and datasets are available at `https://github.com/csjiezhao/AgentTravel`.

## 1 Introduction

The rapid advancement of large language models (LLMs) has opened new opportunities for building agentic intelligent systems in real-world decision-making tasks. Among these, urban travel planning has emerged as a particularly promising and impactful application domain [19, 16]. As a representative case of urban intelligence, travel planning inherently integrates multiple subtasks: retrieving up-to-date information about points of interest (POIs), reasoning over spatial relationships, selecting transportation options, and organizing itineraries that satisfy diverse user preferences and constraints. Such complexity requires LLM-driven systems not only access and integrate heterogeneous knowledge sources, but also demonstrate spatial reasoning and multi-step decision-making capabilities to operate effectively in dynamic urban environments.

Despite recent advances in benchmarking [22], agent architectures [2], and iterative plan refinement [10], several fundamental challenges remain unresolved. First, current LLMs exhibit limited spatial reasoning capabilities—they often fail to accurately account for geographic distances, travel times, or accessibility constraints when generating feasible itineraries [5, 6]. Second, integrating heterogeneous and real-time information from open APIs, transportation platforms, and local knowledge bases remains non-trivial: most existing systems either ignore dynamic contextual factors or depend on narrow, domain-specific data sources. Third, while prior work such as TravelPlanner [19] has proposed evaluation frameworks based on commonsense and hard constraints, there is still a lack of scalable, multi-perspective benchmarks that jointly assess knowledge grounding, contextual reasoning, and the practical quality of generated travel plans.

---

[*]Corresponding author

To address these challenges, we propose **AgentTravel**, a unified framework designed to advance urban travel planning through knowledge-augmented LLM agent. The framework integrates three complementary components designed for reasoning, planning, and evaluation: (1) **TravelLLM**, a domain-adapted base model fine-tuned with curated knowledge about cities, POIs, transportation, and travel constraints. This component enhances the model's spatial reasoning and domain adaptability for diverse urban contexts; (2) **TravelAgent**, an online agentic planner built upon TravelLLM that leverages open Web APIs for real-time information retrieval, maintains structured itinerary memory, and employs adaptive planning strategies to meet user preferences and contextual constraints; (3) **TravelBench**, a scalable benchmark suite with two complementary modules: *KnowEval*, which evaluates factual and spatial knowledge integration using curated urban datasets, and *TripEval*, which measures plan feasibility, personalization, and constraint satisfaction across realistic travel scenarios.

The contributions of this paper are threefold: (1) We release a multi-source urban knowledge dataset covering five representative Chinese cities, encompassing road networks, POIs, attractions, accommodations, and restaurants. The dataset supports both LLM fine-tuning and knowledge-grounded evaluation for urban planning tasks. (2) We develop an online agentic framework that integrates real-time information retrieval, spatially aware planning strategies, and persistent itinerary memory to generate user-centered travel plans. (3) We introduce a comprehensive evaluation suite that jointly assesses knowledge grounding and multi-criteria plan quality, enabling a holistic assessment of knowledge-augmented LLM agents for urban travel planning.

## 2 Related Work

Recent research on LLM-based travel planning [10, 1] can be broadly categorized into two paradigms: **LLM as Planner** and **LLM as Translator**. The former treats the LLM as the central reasoning and generation engine that directly produces travel itineraries, often enhanced with tool use, agent-based strategies, or prompt optimization. The latter leverages the LLM primarily as a natural language interface, translating user requirements into formal or symbolic representations that external solvers can optimize.

**LLM as Planner.** Planner-based approaches focus on empowering LLMs to handle the end-to-end travel planning pipeline, from understanding user constraints to generating detailed itineraries. Early efforts such as TravelPlanner [19] established a benchmark for evaluating an LLM agent's ability to use tools and satisfy commonsense and hard constraints. TravelPlanner+ [14] extended this with personalized user models, highlighting the impact of tailoring itineraries to user preferences. Flex-TravelPlanner [12] examined the robustness of planning under dynamic and uncertain conditions, while NATURAL PLAN [22] revealed persistent challenges in multi-city, long-duration scenarios despite providing full task information. Beyond benchmarking, multi-phase planning frameworks [18] such as TDAG [17] and HyperTree Planning [7] decomposed complex trips into manageable sub-tasks, improving scalability. Additional work has targeted prompt optimization [11, 3], multi-module agent designs such as TravelAgent [2], and dialogue-driven multi-agent planning [21]. Collectively, these studies advance the ability of LLMs to operate as autonomous planners, but most still face limitations in robust spatial reasoning and in integrating diverse real-time data streams into the planning loop.

**LLM as Translator.** Translator-based approaches shift the focus from direct itinerary generation to bridging natural language and structured reasoning systems. In these methods, LLMs convert user queries into machine-interpretable formats—such as symbolic constraint sets, semantic graphs, or formal planning languages—that are then processed by external solvers. For instance, **(author?)** [8] formulated travel planning as a satisfiability modulo theories (SMT) problem, enabling precise constraint handling. ItiNera [15], TRIP-PAL [4], and TTG [9] followed similar pipelines, combining LLM-based parsing with solver-based optimization. ChinaTravel [13] contributed an open benchmark for scalable evaluation of travel planning, focusing on aligning generated plans with real-world travel demands. This paradigm offers strong guarantees on constraint satisfaction and optimality, but often relies on static or incomplete knowledge bases, making it less adaptive to dynamic, multi-source inputs and less capable of leveraging LLMs' generative flexibility for nuanced user preferences.

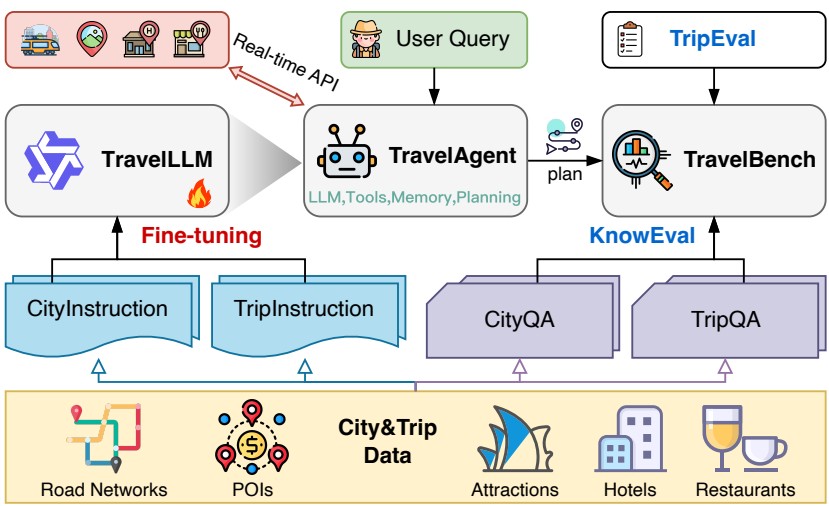

Figure 1: Overview of AgentTravel.

## 3 Preliminaries

**Definition 1 (Urban Travel Plan)** *An urban travel plan $p$ is a structured itinerary spanning $M$ consecutive days for $N$ travelers within an urban environment. It can be represented in a JSON-like format containing fields such as date, attractions, restaurants, accommodations, and transportation, along with optional metadata.*

**Definition 2 (Online Trip Data)** *Online trip data $\mathcal{D}_{\text{on}}$ denotes real-time travel information retrieved from external APIs during planning. It includes attributes of attractions (name, price), restaurants (name, price, cuisine), and accommodations (name, price, hotel type), providing up-to-date references for generating feasible and cost-aware itineraries.*

**Definition 3 (Offline City Data)** *Offline city data $\mathcal{D}_{\text{off}}$ refers to static, city-specific information collected before planning. It comprises road networks, POI datasets, and tourism-related data (e.g., attractions, restaurants, hotels) obtained from public sources. This data serves as a persistent knowledge base that enhances the spatial reasoning and domain knowledge of the underlying LLM.*

**Problem Statement.** Given a user query $q$ in natural language, the goal of urban travel planning is to generate an itinerary $p$ under accessible online data $\mathcal{D}_{\text{on}}$:

$$p = \mathcal{F}(q, \mathcal{D}_{\text{on}})$$

where $\mathcal{F}$ denotes an *agentic planner* built upon LLMs and augmented with offline city data $\mathcal{D}_{\text{off}}$.

## 4 AgentTravel

Figure 1 demonstrates the architecture of *AgentTravel*, which integrates knowledge-augmented modeling, agentic real-time planning, and multi-perspective evaluation. The process starts with a natural-language query, which activates TravelAgent to coordinate interactions among TravelLLM, real-time trip data, and a structured memory that tracks the itinerary in progress. The generated plan is then assessed by TravelBench, which combines KnowEval and TripEval for a comprehensive evaluation of knowledge grounding and planning quality.

### 4.1 TravelLLM

*TravelLLM* is a knowledge-augmented large language model tailored for urban travel planning. We use Qwen 2.5-7B as the backbone and apply Low-Rank Adaptation (LoRA) for efficient domain and spatial knowledge injection. The model is fine-tuned on a hybrid corpus that combines two domain-specific instruction sets—*CityInstruction* and *TripInstruction*—with several open instruction datasets to enhance stability and generalization.

### 4.1.1 CityInstruction: Urban Spatial Knowledge

*CityInstruction* enhances spatial reasoning and geographic understanding using instruction–response pairs derived from curated *offline city data* $\mathcal{D}_{\text{off}}$. It covers two major categories: (1) **Intersection**: mapping intersection names to coordinates (`name2coords`), performing reverse lookups (`coords2name`), and computing distances between intersections (`between_distance`); (2) **Points of Interest (POI)**: linking POI names to addresses (`name2address`) and categories, enabling recognition and reasoning over destinations relevant to travel planning.

### 4.1.2 TripInstruction: Travel-Specific Knowledge

*TripInstruction* focuses on travel-specific entities, enriching the model's understanding of attractions, accommodations, and restaurants to produce realistic and personalized itineraries. It is also derived from $\mathcal{D}_{\text{off}}$ and includes three main categories: (1) **Attractions**: mapping attraction names to addresses (`name2address`), ticket information (`name2ticket`), and opening hours (`name2opentime`); (2) **Hotels**: providing hotel addresses (`name2address`) and average prices (`name2price`) for budget- and location-aware accommodation recommendations; (3) **Restaurants**: linking restaurant names to addresses (`name2address`), price ranges (`name2price`), and cuisine types (`name2cuisine`) for personalized meal planning.

## 4.2 TravelAgent

*TravelAgent* operates through three tightly coupled modules: a structured memory for state tracking, a domain-specific toolbox for real-time data retrieval, and a ReAct-style planning loop for interleaved reasoning and action.

### 4.2.1 Structured Memory for State Tracking

Urban travel planning involves numerous interdependent elements and evolving contextual factors. *TravelAgent* maintains a day-by-day *structured memory* that records itinerary details—attractions, meals, accommodations, transportation, and estimated per-capita costs—providing a persistent state for iterative updates as planning progresses. The schema for each day is defined as:

```
{
    "date": str,
    "num_people": int,
    "visit_attractions": list,
    "breakfast": {"name": str, "cuisines": str},
    "lunch": {"name": str, "cuisines": str},
    "dinner": {"name": str, "cuisines": str},
    "accommodation": {"name": str, "type": str},
    "transportation": {"org-dst": str},
    "cost_per_capita": dict
}
```

### 4.2.2 Domain-Specific Toolbox

The *domain-specific toolbox* is a suite of parameterized functions implemented via JSON-schema-based calls, enabling *TravelAgent* to retrieve, filter, and integrate external travel information during itinerary construction. Each tool serves a specific role in the planning workflow: (1) **MemoryInit**—initializes global trip parameters (e.g., dates, number of travelers) to ensure consistent context for subsequent steps; (2) **AttractionSearch**—queries online sources for detailed attraction information, including names, locations, and attributes; (3) **NearbyRestaurantSearch**—retrieves restaurants within a given radius of a target POI to ensure geographic coherence of meal options; (4) **NearbyHotelSearch**—fetches available accommodations near specified locations for proximity-based lodging selection; (5) **TransportationSearch**—provides feasible routes between two locations to support realistic scheduling and connectivity; (6) **MemoryWrite**—updates the structured memory with newly retrieved or modified itinerary elements, preserving intermediate states; (7) **PlanOutput**—compiles the current memory state into a coherent, user-facing itinerary representation.

### 4.2.3 ReAct-Style Planning Loop

*TravelAgent* follows a ReAct-style planning paradigm [20], interleaving reasoning and tool invocation in an iterative feedback loop. At each iteration, the agent performs three coordinated steps: (1) **State Interpretation**: analyzes the structured memory to evaluate progress and identify missing or inconsistent elements; (2) **Action Selection**: decides between internal reasoning (e.g., sequencing attractions, allocating time slots) and external tool invocation (e.g., querying restaurants, retrieving routes); (3) **State Update**: integrates the results of reasoning or retrieved data into the structured memory, incrementally refining the itinerary state.

## 4.3 TravelBench

### 4.3.1 KnowEval

*KnowEval* assesses an LLM's capability to retrieve and reason over factual urban knowledge before the planning stage. It consists of two complementary subsets: **CityQA**, which focuses on spatial knowledge such as road networks and general POIs, and **TripQA**, which targets domain-specific travel entities including attractions, hotels, and restaurants. Each subset is further structured around fine-grained attribute categories derived from the curated offline dataset $\mathcal{D}_{\text{off}}$.

CityQA covers: (1) *Road attributes* - OD pairs, connectivity, and distances; (2) *POI attributes* - name-to-address mappings. TripQA includes: (1) *Attractions* - address, ticket price, and opening hours; (2) *Hotels* - address and average price; (3) *Restaurants* - address, average price, and cuisine tags. Each knowledge item is converted into a multiple-choice question (MCQ) automatically generated by `GPT-4o-mini` from $\mathcal{D}_{\text{off}}$ and validated by human annotators for factual accuracy and clarity.

### 4.3.2 TripEval

*TripEval* evaluates the *feasibility* and *personalization quality* of travel plans generated by LLM-based agents. It operates on the structured memory produced by the agent and applies a suite of rule-based validators that cross-reference curated POI databases and real-time transportation APIs. The evaluation metrics are grouped into two major categories, as summarized in Table 1.

| Commonsense Constraints | |
|---|---|
| Valid Fields | All required fields in the travel plan are populated. |
| Valid Days | The number of planned days matches the requested trip length. |
| Valid Attractions | Every listed attraction is real and publicly accessible. |
| Valid Restaurants | Every listed restaurant is real and currently operating. |
| Valid Accommodations | All accommodations are valid and bookable. |
| Available Transportation | Transportation between locations is feasible. |
| No Repeated Attractions | No attraction is visited more than once. |
| No Repeated Restaurants | No restaurant is visited more than once. |
| **Preference Constraints** | |
| Reasonable Budget | The total cost remains within the user-specified budget. |
| Favorite Cuisine | The itinerary includes the user's preferred cuisines. |
| Preferred Hotel Type | Accommodation matches the specified hotel category. |

Table 1: Constraint categories in TripEval.

# 5 Experiments

## 5.1 Settings

### 5.1.1 City & Trip Datasets

We construct our datasets from five representative tourist cities in China: Beijing, Shanghai, Guangzhou, Chengdu, and Xi'an. These cities were selected for their combination of rich cultural heritage, diverse urban layouts, and high tourist activity—making them ideal testbeds for

evaluating urban travel planning systems. The *city-level data* is obtained from OpenStreetMap[2] and Amap[3] , providing detailed coverage of road networks, intersections, and POIs. The *trip-level data* is collected from Ctrip[4] , including attractions, accommodations, and restaurants with rich attributes such as prices, operating hours, and category labels. Table 2 summarizes the dataset statistics.

| | City Data | | | Trip Data | | |
|---|---|---|---|---|---|---|
| | Num. Roads | Num. Intersections | Num. POIs | Num. Attractions | Num. Hotels | Num. Restaurants |
| Beijing | 33,794 | 20,327 | 288,852 | 3,471 | 1,473 | 132,379 |
| Shanghai | 38,281 | 18,871 | 424,198 | 3,967 | 1,417 | 117,880 |
| Guangzhou | 25,142 | 17,556 | 483,344 | 3,552 | 1,406 | 82,603 |
| Chengdu | 28,564 | 16,389 | 422,244 | 3,312 | 1,411 | 100,405 |
| Xi'an | 23,176 | 14,215 | 279,080 | 3,107 | 1,439 | 53,263 |

Table 2: Statistics of City and Trip Datasets.

### 5.1.2 Query Generation

To simulate realistic and diverse user requests for itinerary planning, we develop an automated pipeline that generates natural-language queries paired with structured JSON representations. Given a target city and difficulty level, the generator samples key trip parameters - duration, number of travelers, start date, and budget - through controlled randomization. Budgets are derived from a per-capita-per-day baseline cost and adjusted by multiplicative factors for different hotel categories, ensuring internal consistency across trip attributes.

Preference constraints are injected in three tiers: (1) **No preference** - budget constraint only; (2) **Single preference** - one hotel category or one to three preferred cuisines; (3) **Combined preferences** - both hotel category and multiple cuisines. We generate 100 queries per city with difficulty levels, and prompt `GPT-4o-mini` to produce a fluent, user-like query.

### 5.1.3 Metrics

(1) **Delivery Rate (DR)** – the percentage of itineraries successfully completed within the allowed number of reasoning and tool-invocation steps; (2) **Commonsense Pass Rate (CPR)** – the proportion of itineraries satisfying all commonsense constraints defined in *TripEval* (e.g., valid POIs, non-repetition, feasible transportation); (3) **Preference Pass Rate (PPR)** – the proportion satisfying all user-specified preference constraints (e.g., budget, cuisine, accommodation type); (4) **Final Pass Rate (FPR)** – the percentage of itineraries simultaneously meeting both commonsense and preference constraints; (5) **Accuracy (ACC)** – the fraction of correctly answered multiple-choice questions in *KnowEval*, reflecting factual and spatial knowledge grounding.

## 5.2 Results

We evaluate *AgentTravel* against several competitive LLM baselines on both *KnowEval* and *TripEval*. To ensure a fair and controlled comparison, all models operate within the same *TravelAgent* planning framework, sharing an identical prompting template, structured memory schema, ReAct-style reasoning loop, and domain-specific toolbox.

### 5.2.1 Performance on KnowEval

Table 3 reports results on **CityQA** and **TripQA** across five cities. *TravelLLM* ranks first or second in nearly all cases, showing the best overall balance. On **TripQA**, TravelLLM achieves the highest scores in Beijing, Chengdu, and Xi'an, and competitive results in Shanghai and Guangzhou. These gains confirm that domain-specific fine-tuning improves factual recall and reasoning on travel entities. On **CityQA**, GPT-4o-mini leads in Beijing, Shanghai, and Chengdu, while TravelLLM performs better in Guangzhou and Xi'an. This shows that city-level adaptation can match or surpass larger models in localized spatial reasoning.

---

[2]`https://www.openstreetmap.org/`
[3]`https://lbs.amap.com/`
[4]`https://ctrip.com/`

| Model | Beijing (#200) | | Shanghai (#200) | | Guangzhou (#200) | | Chengdu (#200) | | Xi'an (#200) | |
|---|---|---|---|---|---|---|---|---|---|---|
| | CityQA | TripQA | CityQA | TripQA | CityQA | TripQA | CityQA | TripQA | CityQA | TripQA |
| Qwen2.5-7B | 0.420 | 0.580 | 0.445 | **0.645** | 0.475 | **0.655** | 0.475 | 0.515 | 0.450 | 0.585 |
| GLM4-9B | 0.430 | 0.465 | 0.420 | 0.555 | 0.425 | 0.535 | 0.470 | 0.410 | 0.450 | 0.530 |
| Gemma3-12B | 0.325 | 0.490 | 0.420 | 0.475 | 0.330 | 0.455 | 0.435 | 0.455 | 0.390 | 0.550 |
| GPT4o-mini | **0.500** | 0.530 | **0.500** | 0.610 | 0.500 | 0.585 | **0.550** | 0.430 | 0.490 | 0.620 |
| **TravelLLM** | 0.445 | **0.630** | 0.410 | 0.625 | **0.525** | 0.620 | 0.505 | **0.535** | 0.550 | **0.635** |

Table 3: Comparison of different LLMs on KnowEval. Bold denotes the best result, underline denotes the second-best.

| Model | Beijing (#100) | | | | Shanghai (#100) | | | | Guangzhou (#100) | | | | Chengdu (#100) | | | | Xi'an (#100) | | | |
|---|---|---|---|---|---|---|---|---|---|---|---|---|---|---|---|---|---|---|---|---|
| | DR | CPR | PPR | FPR | DR | CPR | PPR | FPR | DR | CPR | PPR | FPR | DR | CPR | PPR | FPR | DR | CPR | PPR | FPR |
| Qwen2.5-7B | 0.97 | 0.18 | 0.43 | 0.15 | 0.89 | 0.12 | 0.25 | 0.04 | 0.91 | 0.11 | 0.18 | 0.00 | 0.94 | 0.18 | 0.47 | 0.11 | 0.90 | 0.19 | 0.53 | 0.19 |
| GLM4-9B | 0.94 | 0.20 | 0.51 | 0.19 | 0.98 | 0.06 | 0.31 | 0.02 | 0.91 | 0.12 | 0.34 | **0.08** | 0.97 | 0.04 | 0.50 | 0.03 | 0.96 | 0.17 | 0.55 | 0.16 |
| Gemma3-12B | 0.29 | 0.00 | 0.17 | 0.00 | 0.34 | 0.00 | 0.15 | 0.00 | 0.31 | 0.00 | 0.14 | 0.00 | 0.31 | 0.02 | 0.09 | 0.01 | 0.13 | 0.00 | 0.07 | 0.00 |
| GPT4o-mini | 1.00 | 0.41 | 0.40 | 0.19 | 1.00 | 0.41 | 0.02 | 0.01 | 1.00 | 0.39 | 0.14 | 0.07 | 1.00 | 0.08 | 0.13 | 0.03 | 1.00 | 0.11 | 0.52 | 0.05 |
| **AgentTravel** | 0.98 | 0.42 | 0.34 | **0.24** | 0.99 | 0.20 | 0.12 | **0.10** | 1.00 | 0.31 | 0.05 | 0.01 | 0.99 | 0.14 | 0.42 | **0.15** | 1.00 | 0.31 | 0.40 | **0.24** |

Table 4: Main results of different LLMs on TripEval. Bold denotes the best result, underline denotes the second-best.

### 5.2.2 Performance on TripEval

Table 4 reports delivery (DR), commonsense (CPR), preference (PPR), and final pass rate (FPR) across five cities. *AgentTravel* achieves near-perfect delivery ($\geq$0.98) across all settings, indicating strong execution stability. GPT-4o-mini performs best on commonsense reasoning, while *AgentTravel* remains competitive in Beijing and Xi'an, outperforming other open models. On personalization, performance is moderate but consistent, slightly below Qwen and GLM in some cities. Notably, *AgentTravel* attains the highest FPR in four cities, reflecting improved overall feasibility.

Despite these advances, LLM-based travel planning remains challenging. Our results suggest that integrating knowledge-grounded reasoning with structured memory offers a promising path toward more reliable and adaptive LLM planners.

## 6 Conclusion

This paper introduced AgentTravel, a unified framework for LLM-based urban travel planning, combining knowledge-grounded modeling, agentic reasoning, and multi-perspective evaluation. Experiments across five Chinese cities show that domain- and city-specific fine-tuning strengthens factual reasoning, while structured agentic planning improves itinerary feasibility. Despite these gains, LLM-based travel planning remains a challenging task, requiring better commonsense reasoning, preference alignment, and adaptability to real-world data.

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
