# A CityInstruction & TripInstruction Examples

*Example* (Intersection-`name2coords`):

```
{
  "instruction": "Please provide the geographical coordinates of a given
      intersection",
  "input": "Zhouzhang Road and Fangyi Road Intersection",
  "output": "115.6906259, 39.5750395"
}
```

*Example* (POI-`name2address`):

```
{
    "instruction": "Please provide the address of a given Point of Interest.",
    "input": "Sanyuan Ecological Park in Beijing",
    "output": "No. 8, Xiaoyunli, Sanyuan Park, Taiyanggong Township, Chaoyang
        District"
}
```

*Example* (Attractions-`name2ticket`):

```
{
    "instruction": "Please tell me the ticket price of a given attraction.",
    "input": "Old Summer Palace in Beijing",
    "output": "The ticket price for the Old Summer Palace in Beijing is 10 CNY."
}
```

*Example* (Restaurants-`name2cuisine`)

```
{
    "instruction": "What is the main cuisine offered by this restaurant?",
    "input": "Zi Shan Restaurant, Mandarin Oriental Wangfujing, Beijing",
    "output": "Zi Shan Restaurant offers ['Cantonese Cuisine', 'Cantonese Dim Sum
        ']."
}
```

# B Prompts

## B.1 ReAct Planning Prompt

```
You are a travel planning assistant. Your task is to help users create detailed
↪   daily travel itineraries (in Chinese) by strictly following the instructions
↪   below.

### Responsibilities
1. Understand user requirements: Accurately extract travel start/end dates, number
↪   of people, budget, preferences, etc.
2. Retrieve information using tools: Use designated tools to gather data on
↪   attractions, restaurants, accommodations, and transportation.
3. Preliminary setup: Before starting the planning task, use `MemoryInit` to
↪   initialize the memory and set up essential information such as travel dates and
↪   group size.
4. Timely record-keeping: Each time a restaurant, accommodation, or transportation
↪   item is obtained, immediately write it to the memory using `MemoryWrite`.
5. Step-by-step itinerary construction: First determine the full list of attractions
↪   to be visited across the trip. Then, collect and record restaurants,
↪   accommodations, and transportation information on a day-by-day basis.
```

```
### Task Execution Flow
#### Phase 1: Plan Attractions Across the Entire Trip
1. Use `MemoryInit` to initialize the memory with travel dates and number of people.
2. Call `AttractionSearch` to retrieve information about attractions in the target
↪  city.
3. Select appropriate attractions and assign them to each day in a balanced manner
↪  (avoid overcrowded schedules).
4. Use `MemoryWrite` to record the attractions for Day 1. Repeat this for each day
↪  until all attractions have been assigned and recorded.

#### Phase 2: Daily Information Collection and Logging
For each day, perform the following steps in sequence:
1. Call `NearbyRestaurantSearch` to obtain breakfast options.
2. Write the breakfast information to the memory using `MemoryWrite`.
3. Repeat the above two steps for lunch.
4. Repeat the above two steps for dinner.
5. Call `NearbyHotelSearch` to find accommodation near the day's attractions.
6. Record accommodation details with `MemoryWrite`.
7. Call `TransportationSearch` to get transportation plans between all visited
↪  attractions for the day.
8. Log the transportation details using `MemoryWrite`.

### Using Thought-Action-Observation Loop
- Thought: Express your current reasoning using natural language. Do not include any
↪  tool calls in this phase.
- Action: Based on your thought, invoke the appropriate tool using valid parameters.
↪  Use the system's function-calling mechanism where possible.
- Observation: Examine the tool's output and use it to guide the next thought.

### Important Guidelines
- Do not use attraction/restaurant/hotel names unless they come from the results
↪  returned by `AttractionSearch` or `NearbySearch`.
- Each piece of information must be collected and recorded **independently**;
↪  merging multiple tasks is not allowed.
- To avoid forgetting data, each collected item must be immediately written to the
↪  memory using `MemoryWrite`.
- For days with multiple attractions, transportation between each pair must be
↪  queried and written separately.
- Each Action phase should involve only one tool invocation for a single specific
↪  task. Multiple tool uses in one action are not allowed.
- After all daily information has been recorded in the memory, call `PlanOutput` to
↪  generate the final complete travel plan.
```

## B.2    Knowledge Evaluation Prompt

```
Here is a multiple-choice question related to urban travel knowledge. You need to
↪  choose the most appropriate answer from A, B, C, and D. Please output only the
↪  letter corresponding to the correct answer, with no additional content.
```

# C    Example Query and Generated Plan

Below is an example of a user query in English and the corresponding structured travel plan produced by our system.

**User Query**

```
I would like a 1-day travel plan in Beijing for 2 people, starting on July 20, 2025,
↪  with a budget of around 2,200 CNY.
```

**Generated Plan**

```
{
    "date": "2025-07-20",
    "num_people": 2,
    "visit_attractions": [
        "Summer Palace",
        "Palace Museum (Forbidden City)",
        "Temple of Heaven"
    ],
    "breakfast": {
        "name": "Palace Museum Restaurant",
        "cuisines": "Chinese"
    },
    "lunch": {
        "name": "Tingliguan Restaurant (Summer Palace Branch)",
        "cuisines": "Chinese"
    },
    "dinner": {
        "name": "Donglaishun Restaurant (Temple of Heaven Branch)",
        "cuisines": "Beijing Cuisine"
    },
    "accommodation": {
        "name": "Atour Light Hotel Beijing Qianmen Temple of Heaven",
        "type": "Comfort"
    },
    "transportation": {
        "Summer Palace → Palace Museum":
        ↪   "From the Summer Palace, walk 791 meters to ...",
        "Palace Museum → Temple of Heaven":
        ↪   "From the Palace Museum, walk 870 meters to ..."
    },
    "cost_per_capita": {
        "Palace Museum": 60,
        "Summer Palace": 30,
        "Temple of Heaven": 10,
        "breakfast": 86,
        "lunch": 153,
        "dinner": 147,
        "accommodation": 300,
        "transit": 8.0
    }
}
```