# OpenReview forum: "[Regular Track]AgentTravel: Knowledge-Augmented LLM Agent Framework for Urban Travel Planning"
_NeurIPS.cc/2025/Workshop_Mexico_City/NORA — NeurIPS 2025 Workshop NORA Poster_

### Official Review · Reviewer_qocC · 2025-11-01

**Rating:** 6
**Confidence:** 3

**Review:**

The paper presents AgentTravel, a framework leveraging large language models (LLMs) for intelligent urban travel planning. Recognizing the complexity of integrating real-time, multi-source data—including points of interest, transportation networks, and user preferences—the authors propose a unified approach that combines knowledge-grounded modeling, agentic reasoning, and multiperspective evaluation. The framework consists of TravelLLM, a domain-adapted model enriched with urban and spatial knowledge; TravelAgent, an agentic planner with structured itinerary memory and real-time data retrieval; and TravelBench, a benchmark designed to assess both knowledge grounding and itinerary quality. Experimental evaluation across five Chinese cities demonstrates that AgentTravel outperforms strong baselines in both factual reasoning and itinerary feasibility, suggesting its potential for enhancing LLM-based urban intelligence systems.

Strengths:

The paper addresses a challenging and highly relevant problem with a novel combination of knowledge-grounded LLMs and agentic reasoning. The integration of domain-specific knowledge into TravelLLM and the structured memory approach in TravelAgent are promising innovations that improve itinerary feasibility. The introduction of TravelBench as a multiperspective evaluation framework demonstrates a thoughtful consideration of both factual correctness and practical usability, which is often overlooked in LLM applications. The experimental evaluation on multiple real-world cities adds credibility to the approach and highlights the potential generalizability of the framework.

Weaknesses:

The paper lacks sufficient technical detail regarding the implementation of both TravelLLM and TravelAgent. Key aspects such as how urban and spatial knowledge is encoded, how the agentic planning mechanism handles dynamic updates or conflicts in real-time data, and the specific structure of the itinerary memory remain unclear. Some aspects of the framework’s integration, such as the interaction between the LLM and real-time data retrieval, are only described at a high level, limiting the reader’s understanding of the underlying mechanisms.

Missing references:

Urbankgent: A unified large language model agent framework for urban knowledge graph construction

KnowAgent: Knowledge-Augmented Planning for LLM-Based Agents

---

### Official Review · Reviewer_StWx · 2025-11-01
**Lack of baseline comparisons and process details**

**Rating:** 4
**Confidence:** 4

**Review:**

Summary:
This paper proposes a unified framework that fine tunes a TravelLLM with CityInstruction and TripInstruction datasets. TravelAgent uses a structured memory for state-tracking, domain-specific tools, and ReAct-style planning. It evaluates on a benchmark constructed against KnowEval and TripEval across 6 evaluation metrics. While the problem is useful, the solution seems to be lacking in novelty, comparison against other LLM-based travel planners, and details on how some of the steps can be replicated on other cities.

Pros:
1. The paper is easy to read.
2. Several evaluation metrics are used.
3. Related work section outlines the area well.

Cons:
1. No comparison against existing LLM-based travel planning baselines mentioned in the related work section.
2. The novelty is unclear. Specifically, what does each component being added to the framework add that doesn't exist in a previous/existing method? This should be emphasized in the writing.
3. Performance is not the best. If GPT-4omini performs as well if not better than TravelLLM and AgentTravel, what is the point of building this framework? What does it add to the outcome that GPT-4omini does not, given that TravelAgent requires fine-tuning and data collection steps?
4. The data collection process for evaluation and fine-tuning is not really described. This is necessary to adapt the framework to other cities. KnowEval and TripEval seem to be constructed by the authors so please describe them in more detail. If not, add references.

Details:
1. Open questions: how scalable is this? This framework needs to be trained on individual cities. What is the effort required to collect the data?
2. What about global travel planning or with more cities in the trip? How would this scale in regard to data collection and training?
3. Can you provide more detail on the domain-specific tool box? Are they non-LLM based?
4. Where does AgentTravel sit in the two LLM-based travel planning paradigms?
5. In table 4 can you include bolds and underlines for all metrics?
6. Missing reference on page 2 (author?)

---

### Official Review · Reviewer_kmUG · 2025-11-04
**TravelAgent - knowledge augmented LLM Agent**

**Rating:** 5
**Confidence:** 4

**Review:**

The paper introduces AgentTravel, a framework for LLM-based urban travel planning that integrates three components:
 (1) TravelLLM, a domain-adapted Qwen 2.5-7B model fine-tuned with urban/spatial knowledge (CityInstruction and TripInstruction datasets derived from offline city data like road networks, POIs, attractions, etc.);
(2) TravelAgent, a ReAct-style agentic planner using structured memory for itinerary tracking, a domain-specific toolbox for real-time API calls (e.g., AttractionSearch, NearbyRestaurantSearch), and iterative reasoning; and
(3) TravelBench, a benchmark with KnowEval (MCQ-based factual/spatial knowledge assessment via CityQA/TripQA) and TripEval (rule-based checks for commonsense/preference constraints in generated plans).
The framework is evaluated on datasets from five Chinese cities (Beijing, Shanghai, Guangzhou, Chengdu, Xi'an), showing improvements in knowledge grounding and plan feasibility over baselines like Qwen2.5-7B, GLM4-9B, Gemma3-12B, and GPT-4o-mini. Code and datasets are released on GitHub.

Key Strengths of the paper -
(1) Practical applicability. Urban travel planning is a timely, real-world use case for LLM agents, blending spatial reasoning, multi-source data integration.
(2) The paper talks about an implemented framework with proper design, including knowledge augmentation (fine-tuning), agentic planning (ReAct loop with memory/toolbox), and evaluation (multi-perspective benchmark).
(3) Dataset Contributions: Releasing multi-source urban data. This may be of use to subsequent research
(4) Strong Evaluation: TravelBench is comprehensive—KnowEval tests grounding pre-planning (MCQs validated by humans), while TripEval uses rule-based metrics (e.g., Valid Attractions, Reasonable Budget) cross-referenced with APIs/databases. Metrics like Final Pass Rate (FPR) holistically capture feasibility. Baselines share the same agent framework for fair comparison.

Weaknesses:
(1) Limited Generalization and Scope: Evaluation is confined to five Chinese cities, relying on region-specific sources (Amap, Ctrip). No cross-cultural validation (e.g., Western cities) or ablation on city diversity (e.g., rural vs. dense urban). Spatial reasoning is presented; however, dynamic edge case evaluation (significant traffic delay or unexpected closure of POI etc.) is not discussed.
(2) Modest Performance Gains: While competitive, results are not transformative—e.g., FPR is highest at 0.24 (Xi'an), and CPR/PPR hover ~0.1–0.4 (Table 4). TravelLLM trails GPT-4o-mini on CityQA in 3/5 cities (Table 3), suggesting fine-tuning on a 7B model doesn't fully close the gap with proprietary LLMs.
(3) Evaluation Limitations: KnowEval MCQs are auto-generated by GPT-4o-mini (potential bias/leakage) and lack open-ended reasoning tests.
(4) Fine-tuning details are sparse (e.g., LoRA hyperparameters, training epochs, loss curves). ReAct loop lacks convergence analysis (e.g., avg. iterations/step limits).
(5) Typos / Omissions - e.g. Authors for Ref [8] is missed out from the text body. Detailed steps of data selection for fine tuning is also absent.

I also do not see any novel algorithm / application (such as custom LoRA variant, RL for planning) being discussed or explored. Results also show modest performance gains. It is unclear how this framework (which is very application specific), can be applied in a broader sense to improve state-of-the-art.

---

### Official Review · Reviewer_1M2Q · 2025-11-06
**More ablation analysis experiments are needed to demonstrate the effectiveness of the method.**

**Rating:** 4
**Confidence:** 4

**Review:**

This paper introduces a knowledge-enhanced large language model (LLM) framework called AgentTravel, designed to improve urban travel planning. The framework generates user-centric travel plans by integrating real-time information retrieval, spatial planning strategies, and persistent itinerary memory. The paper's key contributions include the release of a multi-source city knowledge dataset covering five representative Chinese cities, the development of an online agent framework, and the introduction of a comprehensive evaluation suite to assess the program quality of knowledge-augmented LLM agents.

# Strengths

1.  TravelBench sounds interesting.
2.  The code and dataset are already released, which is potentially helpful to the community.

# Weaknesses

1.  The article mentions that existing LLMs have limited capabilities in spatial reasoning and often cannot accurately consider geographical distance, travel time, or accessibility constraints. However, this article does not explain why LLMs can be equipped with these capabilities through fine-tuning on CityInstruction and TripInstruction.
2.  More ablation analysis experiments are needed to demonstrate the effectiveness of the method.
3.  More details about TravelAgent are needed.  The input and output formats of TravelLLM?

Typo

1.  (author?) in section Related work

---

### Official Review · Reviewer_inbR · 2025-11-06
**Agentic LLM solution to travel planning**

**Rating:** 5
**Confidence:** 4

**Review:**

The paper presents an LLM-Agentic solution to travel planning. the approach consists of three main components, a TravelLLM which is a fine-tuned LLM that is more competent on spatial reasoning and geographical understanding , an LLM TravelAgent that performs the planning and two bench-marking datasets.

Overall the use case is interesting, however, I have the following concerns.

Although datasets and databases regarding POIs, Road Networks, attractions, etc were used for the fine tuning, the element of Knowledge Graphs and Knowledge usage (which is the main topic of the workshop) is missing from the paper. Hence, I am not entirely sure the solution fits within the scope of the workshop.

The technical part is quite abstract and high-level. It would be good to see some more details or elaborate more what does it mean exactly that TravelLLM is more proficient in spatial reasoning. Generally, some running example or more details, e.g., about the CityInstruction and TripInstruction datasets. would be appreciated.

In Page 2 there is an authors citation missing (author?) [8]

---

### Official Review · Reviewer_xE13 · 2025-11-07
**An interesting project that needs some additional technical detail especially on how knowledge graphs are being used**

**Rating:** 5
**Confidence:** 4

**Review:**

The primary strength of the paper is in the combining of data sources with LLM functionality to provide itineraries that use real city information and simulated user preferences. By using information from travel maps, businesses, etc. and comparing that data to user requests, AgentTravel solves a request with a valid itinerary.

This project leans more towards a proof of concept project, since it hasn’t been tried with real user queries or open world options. While the LLMs are used over data including locational data, which is represented in a graph form, it is not clear that any knowledge graph was developed as part of the project itself. It seems to be purely LLMs with RAG and fine tuning over data sources, which are partial (5 large cities). Maybe the authors can clarify this point. There is a comparison of the retrieved data with the artificially generated user preferences and a state update of that structured information, and some of this involves novel constraints in the code that check the structures.

The authors provide a link to the GitHub repository. The code is clearly written and well organized. Reading the paper without looking at the codebase, however, was very confusing. Stylistically, each part of the process is given a name and a short description, which feels like a non-technical code walk through rather than a project report.  However, if the naming is used, it could be carried through the document more helpfully. For example, Figure 1 was good to have, but it might be worth it to make a more involved finer-grained diagram with more of the subsequently named items (in bold in the paper) incorporated so it’s easier to follow how the parts work together.

Given the focus of the NORA workshop on integrating LLMs with knowledge graphs, what I expected to find in this paper was a statement of (1) use of LLMs, including the models being used and the prompts used (provided), and (2) a formal description of the type of knowledge graph being used and information about the decision space over which it operates, the efficiency of that operation, (not provided) and (3) the improvements that the combination of LLMs and knowledge graphs offer over using a series of LLMs with prompts (covered implicitly in the evaluation but could be clearer).

There are some smaller points mentioned in the paper that require explanation in the paper itself.
1. Some additional details on how the queries were generated and what the mapping from queries to the JSON-style files involved (e.g., how were natural language ambiguities handled? Or were they beyond the scope of this work?).
2. For state tracking, this sounds like system-internal state tracking as it gathers all of the needed information before returning a valid itinerary. It would be good to explain this since, in such travel systems, in actual practice, there is often a state with respect to the user’s requests and it can be iterated with their feedback.
3. Know Eval, which occurs on the edge from the QA data to Travel Bench in figure 1 uses data about the trip and city to check the validity. The data is said to be LLM generated and checked by human annotators, but no details are given at all about this annotation.
4. There are no examples of what the final output looks like in the paper. It would be good to show  an actual itinerary solution along with the originally stated user preferences.
5. Ideally, the time it takes to go from NL constraints to a valid itinerary solution would be given with some analysis of queries that led to more latency.
6. It would help, when mentioning portions that used LLM prompts, to let readers know that the prompts are in the prompts.py file in the repo, as many papers put them in the appendix.

Whether the issues that cause industry bottlenecks with this type of project are to be alleviated by this approach is not clear from the presentation. The plan for scaling to more open data or to the variety of user requests and phrasings that will occur is hard to discern. Suggestion (5) above would be a good place to start in making a case for scalability.

---

### Official Review · Reviewer_UnrT · 2025-11-07
**Review of AgentTravel: Knowledge-Augmented LLM Agent Framework for Urban Travel Planning - Weak Accept**

**Rating:** 6
**Confidence:** 5

**Review:**

This paper proposes AgentTravel, a unified framework for urban travel planning that integrates
(1) a domain-adapted LLM (TravelLLM),
(2) an agentic planning component that leverages structured memory and tool-based real-time retrieval (TravelAgent), and
(3) a comprehensive evaluation benchmark assessing both knowledge grounding and itinerary feasibility (TravelBench)
Experiments on five Chinese cities demonstrate improved spatial reasoning and more feasible itineraries compared to several LLM baselines.

1. Quality
The paper is generally well-executed and offers a complete system: datasets, models, agent execution loop, and evaluation benchmark. The system design is coherent and components are explained in detail. Experimental comparisons are fairly thorough, covering both knowledge grounding and real-world planning constraints. However, certain result patterns are uneven—e.g., performance varies significantly across cities and evaluation dimensions, and some improvements are modest.

2. Clarity
The paper is mostly clear and readable. The system architecture (Fig. 1) is helpful, and definitions and benchmarks are explained clearly. Nonetheless, some parts (particularly TripEval) rely heavily on implementation details that are not fully described, and the narrative occasionally blends contributions and results without sufficiently isolating key insights.

3. Originality
Most system elements have conceptual overlap with prior agent-based planners (e.g., TravelPlanner, CityGPT, Itinera). The main novelty lies in the integration of a domain-adapted LLM with structured memory and real-time tool use and a multi-level benchmark evaluating both knowledge grounding and itinerary feasibility.

4. Significance
Urban travel planning is a relevant and impactful application domain. The released datasets, code, and evaluation suite enhance practical research value. The benchmark component, in particular, may become a useful resource for future work. However, the paper does not fully address scalability to:
(1) Cross-city generalization beyond the five selected cities,
(2) Multi-day, multi-city trips with inter-city transportation,
(3) Real-time data volatility and error propagation.


Pros:
(1) Comprehensive system that includes model, agent, and evaluation suite.
(2) Domain-specific fine-tuning improves spatial and POI knowledge grounding.
(3) Structured itinerary memory offers a clear improvement over naive generation.
(4) Evaluation benchmark is thoughtfully designed and likely reusable.
(5) Strong delivery and feasibility performance on several test cities.

Cons:
(1) Originality is moderate; many components resemble adaptations of existing planning-agent frameworks rather than introducing fundamentally new techniques.
(2) Preference satisfaction performance is still low, suggesting the system struggles with personalization.
(3) Limited scalability discussion: unclear how well TravelLLM generalizes outside the selected cities or to unseen travel contexts.
(4) Evaluation focuses heavily on rule-based constraints, but not on user satisfaction or trip quality as experienced in real-world scenarios.
(5) Ablation analysis is limited, it is not entirely clear how much each component contributes individually.

I have a few suggestions:
(1) Could the authors provide ablation studies isolating the contribution of TravelLLM vs. TravelAgent memory vs. benchmarks?
(2) How does the system handle noisy, missing, or conflicting online API data?
(3) Could preference alignment be improved through reinforcement learning or retrieval-augmented personalization models?
(4) Consider including evaluation of multi-city travel, which reflects many real travel scenarios.